# Towards Perfect Ultra-Broadband Absorbers, Ultra-Narrow Waveguides, and Ultra-Small Cavities at Optical Frequencies

**DOI:** 10.3390/nano12132132

**Published:** 2022-06-21

**Authors:** Kiyanoush Goudarzi, Moonjoo Lee

**Affiliations:** Department of Electrical Engineering, Pohang University of Science and Technology (POSTECH), Pohang 37673, Korea

**Keywords:** metallic photonic crystals, position disordering, radius disordering

## Abstract

In this study, we design ultra-broadband optical absorbers, ultra-narrow optical waveguides, and ultra-small optical cavities comprising two-dimensional metallic photonic crystals that tolerate fabrication imperfections such as position and radius disorderings. The absorbers containing gold rods show an absorption amplitude of more than 90% under 54% position disordering at 200<λ<530 nm. The absorbers containing silver rods show an absorptance of more than 90% under 54% position disordering at 200<λ<400 nm. B-type straight waveguides that contain four rows of silver rods exposed to air reveal normalized transmittances of 75% and 76% under 32% position and 60% radius disorderings, respectively. B-type L-shaped waveguides containing four rows of silver rods show 76% and 90% normalized transmittances under 32% position and 40% radius disorderings, respectively. B-type cavities containing two rings of silver rods reveal 70% and 80% normalized quality factors under 32% position and 60% radius disorderings, respectively.

## 1. Introduction

Metallic photonic crystals (MPHCs) have brought a new era for manipulating electromagnetic waves at the nanoscale. They are periodic metallic structures in which permittivity modulation occurs in one (grating) [1,2], two [3,4,5], and three dimensions [6,7,8]. These structures have the merits of low costs, easy fabrication, and small sizes, making them candidates for application in optical integrated circuits. The constitutive parts of the metallic structures are mainly gold (Au) and silver (Ag) that exhibit both negative and positive real permittivity, showing extraordinary responses to electromagnetic waves [9,10,11]. The materials have a high carrier concentration and low mobility; this entails a substantially negative real permittivity and material loss [12], which are accountable for reflectance and absorption, respectively. Due to the reflection, a metallic photonic bandgap (PBG), a frequency region that acts as a mirror to block electromagnetic waves, appears in MPHCs [7,13,14,15]. Several crucial optical components have been designed, such as filters [16,17], waveguides [18,19], and cavities [20,21], using PBGs. Based on the absorption in the metallic materials, several optical absorbers have been designed [22,23,24]. The metallic absorbers operating over ultra-violet (UV) and visible wavelengths are mainly applicable in photodetectors, control of thermal radiations, and photovoltaics. Owing to the reflection property of MPHCs, they are crucial candidates in designing waveguides and cavities. Waveguides are the key components in photonic devices that interconnect sources and detectors, called nanocircuits. They have proven their effectiveness over the visible spectrum in mode conversion [25], multimode polarization manipulation [26], coherent control of nanoscale directional couplers [27], and circuits for visible quantum emitters [28]. Optical cavities localize and enhance light beyond the diffraction limit and within a small volume. These components are vastly utilized in various nanolasers such as spasers [29,30], Fabry–Perot lasers [31,32], and whispering-gallery-resonator-based lasers [33,34].

Reducing the size of MPHCs for subwavelength operation (corresponding to the visible and UV spectra) brings a new challenge of fabrication imperfections [2,5,7,8,13,15,35] for optical components, such as absorbers, waveguides, and cavities. The imperfections appear as position and radius disorderings that influence the functionality of the structures [13,15,36,37,38]. As a result, designing ultra-broadband absorbers for UV and visible light, ultra-narrow and highly efficient waveguides for visible light, and ultra-small and high-quality-factor cavities for visible light that tolerate fabrication imperfections is of great importance.

This research theoretically studies the effects of fabrication imperfections on absorbers, waveguides, and cavities over the interval spectrum of 200<λ<600 nm. The optical components are designed using 2D MPHCs. The 2D MPHCs contain long Au or Ag cylinders in periodic arrangements of cubic and hexagonal patterns in air. Ultra-narrow waveguides and ultra-small cavities are designed using two rows of cylindrical Ag exposed to air, which tolerate fabrication imperfections.

The paper contains three sections. The second section describes two types of fabrication imperfections for the 2D MPHCs that elucidate the design procedure of absorbers with tolerance to fabrication imperfections; also it describes the design of straight and L-shaped waveguides and cavities under position and radius disorderings. The last part provides conclusions.

## 2. The Effects of Disorderings

In the subwavelength regime, the effect of fabrication imperfections is inevitable in 2D MPHCs. Fabrication imperfections are categorized as position and radius disorderings. By introducing $x^{i}=x_{0}^{i}+\sigma_{p}U_{x}$ and $y^{i}=y_{0}^{i}+\sigma_{p}U_{y}$ as the position of a rod, the position disordering parameter is introduced as $\eta_{p}=\frac{\sigma_{p}}{a}$, where $\left(x_{0}^{i},y_{0}^{i}\right)$ is the origin position, σp is the strength of disordering, *a* is the lattice constant, and $U_{i}$(i:x,y) is a random variable over an interval [−1, 1] [36,37,38,39]. For the radius disordering, the radius of a rod is $R^{i}=R_{i}^{0}+\sigma_{r}U_{r}$, where $R_{i}^{0}$ is the origin position of the rod, $\sigma_{r}$ is the strength of the radius disordering, and $U_{r}$ is a random variable over the interval [−1, 1]. The radius-disordering parameter is $\eta_{r}=\frac{\sigma_{r}}{R_{i}^{0}}$ [39]. The effect of disordering is used in designing ultra-broadband optical absorbers, ultra-narrow waveguides, and ultra-small cavities that tolerate position and radius disorderings to some extent. The 2D finite-difference time-domain (FDTD) module of Lumerical software is utilized to numerically calculate derived Maxwell’s equations on the 2D MPHC structures.

### 2.1. Ultra-Broadband Optical Absorbers

The structure of the 2D MPHC absorbers consists of an array of 30×30 cylindrical metallic rods (either Au [40] or Ag [41]) exposed to air in a cubic pattern (Figure 1a,d). The radius and lattice constant of the structures are *r* = 15 nm and *a* = 130 nm, respectively. The 2D MPHCs under position disordering at $\eta_{p}=38\%$ and radius disordering at $\eta_{r}=53\%$ are shown in Figure 1a,d, respectively. Transverse electric (TE) polarized (electric field along the *x*-direction) plane waves are illuminated from the bottom and recorded from the top of the structures.

The simulation result for a bulk Au reveals the absorption amplitude of 0.64 over a wavelength range of 280–480 nm, while the absorption amplitude sharply decreases at λ>480 nm (Figure 1b,e); solid black line). The absorption for the bulk Au is due to the higher imaginary permittivity over the wavelength range, and the sharp decrease is due to lower imaginary permittivity and higher negative real permittivity (higher reflection) (Figure 2b). Shining a plane wave upon the 2D MPHCs containing 30×30 Au cylinders exposed to air at $\eta_{p}=0\%$ (no position disordering) reveals an absorption of more than 0.95 over 300<λ<530 nm and a dip at $\lambda_{B}=260$ nm, which shows the effect of periodicity and is called the Bragg bandgap. At $\lambda_{B}$, the input wave is reflected, and the intrinsic absorption limits the reflection amplitude owing to the imaginary permittivity. In contrast to the bulk gold, the 2D MPHC reveals higher and broadband absorption owing to the increased surface of the structure and enhanced interaction of light with matter. Increasing the position disordering of $\eta_{p}=0\%$ to $\eta_{p}=54\%$, increases the bandwidth of absorption. The broad bandwidth absorption results from the disappearing of the Bragg dip. Bragg wavelength obeys λ∝acos(θ), where θ is the propagation angle, which strongly depends on the lattice constant and the angle of propagation. Increasing the position disordering deteriorates *a* and θ; as a result, the Bragg wavelength disappears (Figure 1b,c). Although increasing the position disordering increases the absorption bandwidth, increasing the radius disordering does not change the periodicity; as a result, the Bragg wavelength and the absorption bandwidth remain unaffected (Figure 1e,f).

Bulk silver under the TE polarized plane wave illumination reveals an absorption of approximately 0.73 over 200<λ<280 nm that is due to the higher imaginary permittivity (Figure 2b; dot-dashed red curve). An absorption peak of 0.9 at λ=300 nm is due to the positive real permittivity peak at the wavelength, which suppresses the reflection (Figure 2b; dotted blue curve). At λ>300 nm, the absorption decreases sharply, owing to the lower imaginary permittivity and higher negative real permittivity, that results in increasing the reflection. Illuminating the plane wave upon the 2D MPHC containing Ag components at $\eta_{r}=\eta_{p}=0\%$ (no radius and position disorderings) reveals absorption spectra with two dips. The first dip at $\lambda_{B}=260$ nm is a Bragg wavelength that shows the reflection, and the other occurs at $\lambda_{M}=350$ nm owing to Mie scattering. By increasing the position disordering from $\eta_{p}=0\%$ to $\eta_{p}=54\%$, the Bragg and Mie dips disappear, resulting in a broad bandwidth with an absorption amplitude A>90%. In contrast to the position disordering, by increasing the radius disordering from $\eta_{r}=0\%$ to $\eta_{r}=80\%$, the absorption spectra remain unchanged due to the preserving of lattice constant and propagation angle. Owing to the low negative real and positive imaginary permittivity (Figure 2b; solid blue and dashed red curves), the Mie wavelength shows a weak dip and is mainly affected by position disordering (Figure 1b,c).

The photonic band structure explains the absorbance for the 2D MPHCs. In this study, as an example, the band structure of a 2D MPHC containing Au rods exposed to air under exposure to TE polarized waves is calculated. The imaginary permittivity of the metallic rods causes fast decaying in electromagnetic fields and an inaccurate band structure; as a result, for calculation of the band structure, the imaginary permittivity of Au is considered to be zero. The band structure contains numerous flat bands over high absorption spectra approximating the surface-plasmon frequency $\omega_{\mathrm{sp}}=\frac{\omega_{p}}{\sqrt{2}}$ (Figure 2a). $\omega_{p}$ for Au is $1.37\times 10^{16}$$\left(\frac{\mathrm{rad}}{s}\right)$ [42]. The flat bands reveal a slow-light regime that results in the high coupling of light to the MPHCs. The physics behind the absorption of the 2D MPHC absorbers originate from impedance matching between the structure and air. For the structure, the effective permittivity and permeability are defined as $\tilde{\varepsilon }\left(\omega \right)=\varepsilon_{1}\left(\omega \right)+i\varepsilon_{2}\left(\omega \right)$ and $\tilde{\mu }\left(\omega \right)=\mu_{1}\left(\omega \right)+i\mu_{2}\left(\omega \right)$, respectively, as a result, the effective refractive index of the structure is $\tilde{n}\left(\omega \right)=\sqrt{\frac{\tilde{\varepsilon }\left(\omega \right)}{\tilde{\mu }\left(\omega \right)}}$. The impedance matching between the structure and air happens when the refractive index of the structure is approximately the same as the refractive index of air (n=1). Under the impedance matching condition, $\tilde{\varepsilon }\left(\omega \right)=\tilde{\mu }\left(\omega \right)$, that reveals the perfect absorption [43,44].

The electrical field distributions of the absorbers containing either Au or Ag cylindrical rods under $\eta_{p}=53\%$ at $\lambda =\lambda_{B}$ reveal the penetration of electromagnetic waves into the structure, resulting in high absorption of the waves (Figure 3a,b); the electrical field distributions of the structure under $\eta_{r}=80\%$ represent the penetration of electromagnetic waves into the structure, indicating the absorption of the waves, which is due to the loss of the metallic rods and also the distributions reveal the reflection of the waves owing to the periodicity of the structure (Figure 3c,d). The comparisons between the previous studies and the proposed absorber are represented in Table 1. Although References [45,46,47] show high average absorption, they do not cover the ultra-violet spectrum. Additionally, their high average absorption is due to the high extinction coefficient of Tungsten. Ref. [48] covers the ultra-violet spectrum, but it suffers from low average absorption. Our structure reveals the highest average absorption that also covers the ultra-violet spectrum. The other merit of our proposed absorber is immune to fabrication disorderings; as a result, it shows the lack of dependency on the angle of incident light.

### 2.2. Designing Optical Waveguides and Cavities

Here, 2D MPHC waveguides consisting of 15×15 Ag rods exposed to air under transverse magnetic (TM) polarized light illumination are designed. The structural parameters are *r* = 50 nm and *a* = 160 nm. Ag has a lower and constant imaginary permittivity and higher negative real permittivity over 340<λ<600 nm, which results in low absorption and high reflection (Figure 2). The reflection region reveals a metallic band gap, which can be explained by the bond theory. The Bond theory for 2D MPHCs assumes an optical state in the form of a quasi-bond state for each rod. The quasi-bond state for each rod couples to its adjacent rod and create a PBG between quasi-bond states [15]. Owing to the high skin depth of Ag and Au at 200<λ<600 nm, each rod exhibits optical states, and the bond theory is applicable in the case of the 2D MPHCs (Appendix A).

Electrical field distributions show electric hotspots between the rods along the *x*-direction under a TE polarized plane wave (Appendix A) and no electric hotspots under the illumination of a TM polarized plane wave (Appendix A). Under TE circumstances, the incident wave causes the creation of electric dipoles (**P**) in each rod and results in coupling between them. Under TM conditions, the electric dipoles do not couple, and the hotspots do not appear between the Ag rods. Under the TM polarized incident light wave, the 2D MPHC structure behaves as a mirror that blocks electromagnetic waves and is called a metallic bandgap that is suitable for designing optical components.

The absorption and reflection spectra of the 2D MPHCs containing 15×15 Ag rods exposed to air show robustness to the position disordering at $\eta_{p}=44\%$ and the radius disordering at $\eta_{r}=60\%$ (Figure 4). Owing to the high imaginary permittivity, the absorption is high at 200<λ<390 nm, and owing to the low imaginary permittivity and high negative real permittivity, absorption is less, and reflection is high at 390<λ<600 nm. The first reason behind the robustness of the absorption and reflection spectra to high position and radius disorderings is the high imaginary permittivity and high negative real permittivity of Ag, respectively (Figure 2b). The second reason is the creation of quasi-bond states due to the penetration of electromagnetic waves in Ag (skin depth) and the coupling of the quasi-bond states that causes a robust bandgap under disordering. Owing to the structural parameters and the TM polarized incident plane wave, none of the Bragg and Mie dips appear in the absorption and reflection spectra (Figure 4). Furthermore, in the incident TM polarized wave (electric field is along the long axis of rods), the surface plasmons are not excited. By exploiting the robustness of the reflection to the position and radius disorderings over 390<λ<600 nm, the straight and L-shaped waveguides and cavities are designed.

#### 2.2.1. Ultra-Narrow Straight and L-Shaped Optical Waveguides

Three types of straight and L-shaped waveguides are designed using the 2D MPHCs containing 15×15 Ag rods exposed to air. Creating straight or L-shaped defects in the structure localizes a guided mode inside the reflection spectra 390<λ<600 nm (Appendix A). The guided modes propagate inside the defects with high transmission efficiency. The straight (Figure 5a,d) and L-shaped (Figure 5g,j) defects inside the structure are called A-type structures. The straight (Figure 5b,e) and L-shaped (Figure 5h,k) waveguides with four rows of Ag rods exposed to air are called B-type structures. The straight (Figure 5c,f) and L-shaped (Figure 5i,l) waveguides with two rows of Ag rods exposed to air are called C-type structures. The incident waves are TM polarized Gaussian in orange, illuminated at the bottom of the waveguides, which are propagated from −y to *y* and recorded at the top for the straight and the right for the L-shaped waveguides, respectively (Figure 5).

The normalized transmission amplitudes ($\frac{T}{T_{0}}$) of the straight waveguides are high and approximately the same for A and B with $\frac{T}{T_{0}}>0.78$ at $\eta_{p}=25\%$ and $\frac{T}{T_{0}}>0.84$ at $\eta_{r}=40\%$, respectively. Additionally, they reveal $\frac{T}{T_{0}}>0.75$ at $\eta_{p}=32\%$ and $\frac{T}{T_{0}}>0.76$ at $\eta_{r}=60\%$ ((Figure 6a,e); solid black and dashed blue curves, respectively). The normalized transmission amplitudes under the radius disordering for the A and B structures are higher than those under the position disordering. Higher normalized transmission amplitudes are due to the preserving periodicity of the structure. The normalized transmissions for C-type straight waveguides are lower than A and B, originating from fewer rods. Owning to the large distances created between adjacent rods by the breaking symmetry of the position disordering, the normalized transmission amplitude is less than that in the case with radius disordering (Figure 6a,e). As noticeable from the logarithmic electromagnetic power distributions at λ=490 nm, the A and B-type straight waveguides at $\eta_{p}=19\%$ show high-confinement of electromagnetic waves (Figure 6b,c) and escaping electromagnetic waves for the C-type waveguide (Figure 6d). Additionally, the A and B-type waveguides at $\eta_{r}=24\%$ shows high confinement of light due to the preserved periodicity of the structure (Figure 6f,g); the C-type structure at $\eta_{r}=24\%$ shows the escape of electromagnetic waves to air (Figure 6h).

The normalized transmission amplitudes for the L-shaped waveguides are high enough and approximately the same for A- and B-type waveguides under position and radius disorderings ((Figure 7a,e); solid black and dashed blue curves, respectively). The A and B types tolerate the position disordering at $\eta_{p}=32\%$ with $\frac{T}{T_{0}}>0.76$, and the waveguides tolerate the radius disordering at $\eta_{r}=40\%$ with $\frac{T}{T_{0}}>0.9$. As evident, reducing the number of rows from 14 (A-type) to 4 (B-type) does not change the normalized transmission amplitude under the position and radius disorderings at $\eta_{p}=25\%$ and $\eta_{r}=40\%$. $\frac{T}{T_{0}}$ decreases at higher position and radius disorderings owing to the greater distances between the rods (Figure 7a,e). $\frac{T}{T_{0}}>0.33$ for C-type L-shaped waveguide at $\eta_{p}=32\%$ and $\frac{T}{T_{0}}>0.42$ for the waveguide at $\eta_{r}=60\%$. The logarithmic electromagnetic power distributions for the A- and B-type L-shaped waveguides at λ=490 nm and under either $\eta_{p}=19\%$ (Figure 7b,c) or $\eta_{r}=24\%$ (Figure 7f,g) reveal that a sharp $90^{\circ }$ bend does not change the guided mode to radiation modes, and the electromagnetic mode is preserved as a guided mode. Increasing the distance between the rods due to either the position or radius disorderings decreases the quasi-bond modes coupling between the rods and reduces the transmission amplitudes and results in fleeing electromagnetic waves (Figure 7d,h). The average transmissions of straight and L-shaped waveguides versus position and radius disorderings are summerized in Table 2 and Table 3, respectively. Owing to the imaginary permittivity of silver over the metallic band, increasing the length of the straight and L-shaped waveguides exponentially decreases the normalized transmittance. For the B-type straight and L-shaped waveguides, the normalized transmittance follows the exponential functions of $\frac{T}{T_{0}}=e^{-0.015N}$ and $\frac{T}{T_{0}}=e^{-0.0135N}$, respectively (where *N* = 1, 2, ⋯ is the number of removed silver rods).

#### 2.2.2. Ultra-Small Cavities

In this part, the cavities are created by removing a rod (point defect) from the 2D MPHCs that contain Ag rods in a hexagonal pattern. Making a point defect, localizes a high-quality mode over the metallic bandgap (0.39<λ<0.6) and is used to design the cavities (Appendix A). Three cavities are introduced regarding the number of surrounded Ag rods as a ring from the point defect. A point defect surrounded by eight, two, and one rings of Ag rods are called A- (Figure 8a,d), B- (Figure 8b,e), and C- (Figure 8c,f) type cavities, respectively. The TM polarized dipole sources are placed in the defect area where the light is recorded using point monitors.

The average normalized quality factor ($\frac{Q}{Q_{0}}$) for the A-type cavities is more than 1 at the position disordering $\eta_{p}=32\%$, indicating an increase in the constructive interference of light due to the interaction of light by several rings of Ag rods (Figure 9a; solid black curve). Additionally, the quality factor variance increases with increasing the position disordering owing to the escaping electromagnetic waves between the rods. The B-type cavities has higher variations of the normalized quality factor under the position disordering due to the escaping electromagnetic waves over two rings of Ag rods. For the A-type cavities, $\frac{Q}{Q_{0}}>0.93$ at the radius disordering $\eta_{r}=40\%$, and $\frac{Q}{Q_{0}}>0.75$ for B-type cavities at $\eta_{r}=40\%$ (Figure 9e). The A- and B-type cavities show lower variations under the radius disordering owing to the preservation of the periodicity of the structure. The C-type cavities show approximately the least constant normalized quality factor under the position and radius disorderings (Figure 9a,e; dot-dashed red curves). Moreover, the C-type cavities containing one ring of Ag rods cannot realize the high confinement of the light wave in the defect due to large distances between the rods. The normalized transmissions versus position and radius disorderings for the 2D MPHC cavities are summarized in Table 4. The logarithmic electromagnetic power distributions for the A- and B-type cavities at λ=428 and 400 nm under $\eta_{p}=19\%$ (Figure 9b,c) and at λ=424 and 410 nm under $\eta_{r}=24\%$ (Figure 9f,g) reveal the high confinement of the electromagnetic waves. Additionally, the C-type cavities at λ=400 nm under $\eta_{p}=19\%$ (Figure 9d) and at λ=410 nm under $\eta_{r}=24\%$ (Figure 9h) show the confinement of the electromagnetic waves at the center and escaping of electromagnetic waves owning to the large distances between rods. Owing to the high negative real permittivity of the Ag rods, the position and radius disorderings apply a phase difference, resulting in variations in the frequency of the guided mode under the straight and L-shaped defects and confined modes in the cavities.

## 3. Conclusions

This paper theoretically proposed three optical components, namely absorbers, waveguides, and cavities. The components contain 2D metallic (Au and Ag for absorbers and Ag for waveguides and cavities) rods exposed to air. The absorbers consist of 2D MPHC of gold rods in air represented A≥0.9 under $\eta_{p}=54\%$ over the 200<λ<530 nm. The 2D FDTD simulation results for the A-type straight waveguides showed $\frac{T}{T_{0}}\geq 0.77$ and 0.78 under $\eta_{p}=32\%$ and $\eta_{r}=60\%$, respectively, also, for A-type L-shaped waveguides $\frac{T}{T_{0}}\geq 0.8$ and 0.79 under $\eta_{p}=32\%$ and $\eta_{r}=60\%$, respectively. The simulation results for the B-type straight waveguides revealed $\frac{T}{T_{0}}\geq 0.78$ and 0.85 under $\eta_{p}=25\%$ and $\eta_{r}=40\%$, respectively. The results for the B-type L-shaped waveguides represented $\frac{T}{T_{0}}\geq 0.80$ and 0.92 under $\eta_{p}=25\%$ and $\eta_{r}=40\%$, respectively. The B-type cavities reveal $\frac{Q}{Q_{0}}\geq 0.85$ and 0.8 under $\eta_{p}=32\%$ and $\eta_{r}=60\%$, respectively. The designed ultra-broadband absorbers are potential components for achieving photodetectors, thermal radiation control, and photovoltaics. The ultra-narrow waveguides are crucial components for nanocircuits in polarization manipulation, control coherent, quantum emitters, and color routers for complementary metal-oxide-semiconductor image sensors. The ultra-small cavities are potential candidates for application in high-density integrated circuits and nanolasers.

## Figures and Tables

**Figure 1 nanomaterials-12-02132-f001:**
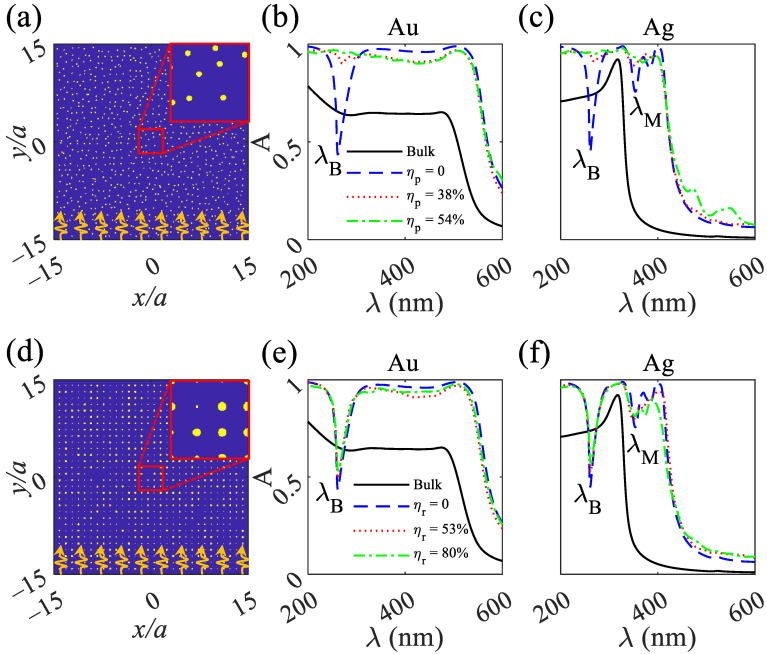
(**a**) Schematic of 2D MPHC absorbers consist of 30×30 cylindrical metallic rods in air at $\eta_{p}=38\%$. (**b**,**c**) show absorption spectra for Au and Ag, respectively, under $\eta_{p}$ = 0, 38, and 54%. (**d**) Schematic of 2D MPHC absorbers consist of 30×30 cylindrical metallic rods in air at $\eta_{r}=53\%$. (**e**,**f**) show absorption spectra for Au and Ag, respectively, under $\eta_{r}$ = 0, 53, and 80%. The insets in (**a**,**d**) show the magnification of 4×4 cylindrical rods in air. The yellow color shows the cylindrical rods, either Au or Ag, and the blue color shows air. TE polarized plane waves are depicted in an orange color and located at the bottom of (**a**,**d**) and they propagate from −y to *y* directions and recorded at the top of the structures. $\lambda_{B}$ and $\lambda_{M}$ stand for Bragg and Mie wavelengths, respectively.

**Figure 2 nanomaterials-12-02132-f002:**
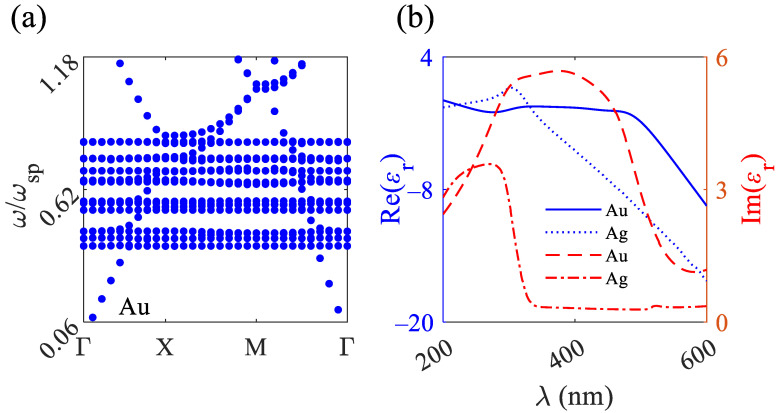
(**a**) Band structure of 2D MPHCs consists of Au rods in air under TE polarized light. (**b**) The real and imaginary parts of permittivity for Au and Ag over a wavelength range of 200–600 nm.

**Figure 3 nanomaterials-12-02132-f003:**
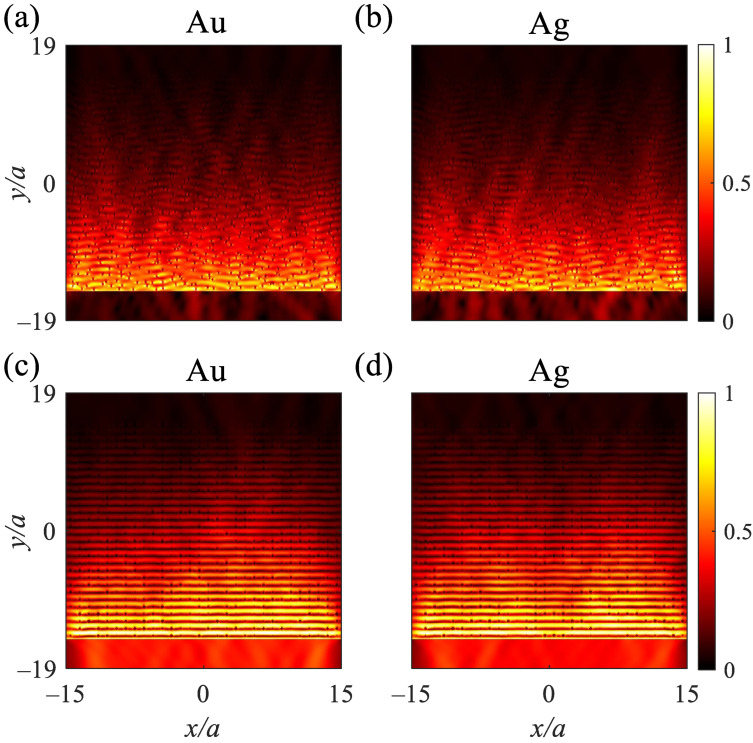
Normalized electrical field distribution of the absorber containing 30×30 cylindrical (**a**) Au and (**b**) Ag rods in air under $\eta_{p}=53\%$, respectively. Normalized electrical field distribution of the absorber under $\eta_{r}=80\%$ containing cylindrical (**c**) Au and (**d**) Ag rods in air, respectively. The electrical field distribution are recorded at $\lambda =\lambda_{B}$; TE polarized plane waves are located at the bottom of the structure and propagate from −y to *y* directions and recorded at the top of the structures.

**Figure 4 nanomaterials-12-02132-f004:**
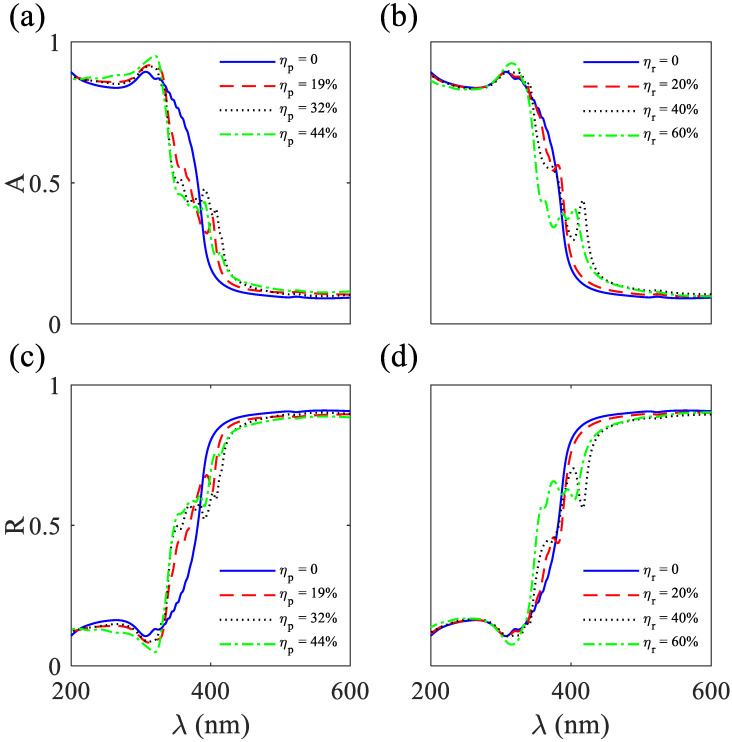
Absorption (A) spectra for 2D MPHCs consist of 15×15 Ag rods in air under TM polarized illuminations under (**a**) position and (**b**) radius disorderings. Reflection (R) spectra of the 2D MPHCs consisting 15×15 Ag rods in air under TM polarized illuminations under (**c**) position and (**d**) radius disorderings.

**Figure 5 nanomaterials-12-02132-f005:**
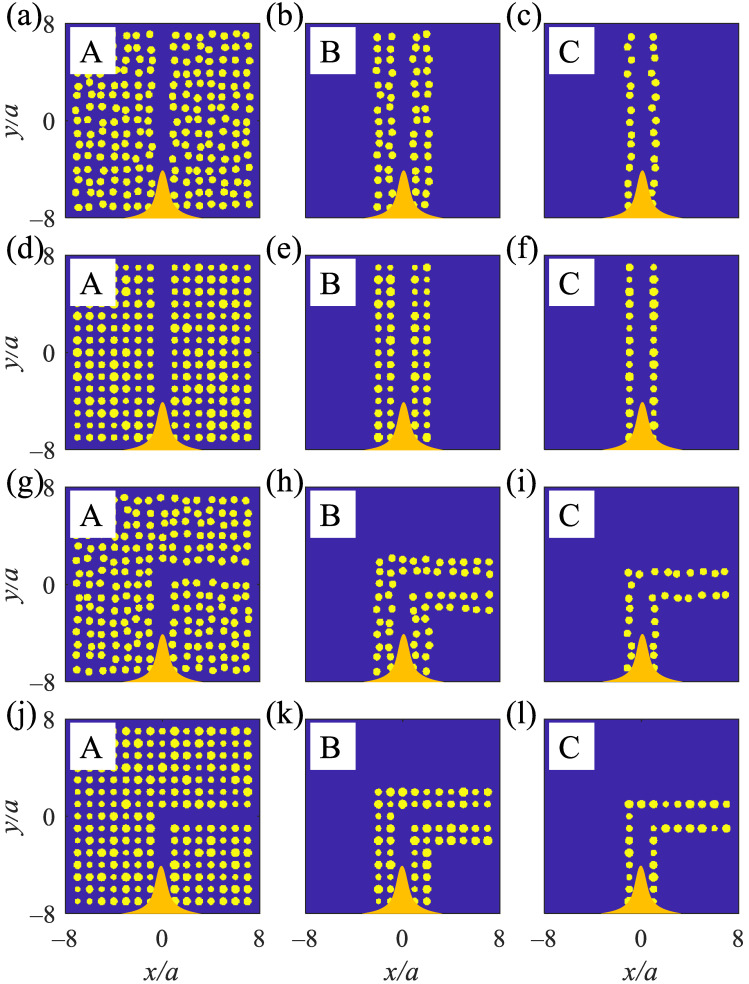
2D MPHCwaveguides contain Ag rods in air. (**a**–**f**) show straight waveguides of A, B, and C at $\eta_{p}=19\%$ and $\eta_{r}=24\%$, respectively. (**g**–**l**) show A, B, and C-type L-shaped waveguides at $\eta_{p}=19\%$ and $\eta_{r}=24\%$, respectively. At the bottom of the waveguides, the orange bell shapes are Gaussian waves that propagate from −*y* to *y* and are recorded at the top and right of the straight and L-shaped waveguides, respectively—yellow: Ag rods and blue: air.

**Figure 6 nanomaterials-12-02132-f006:**
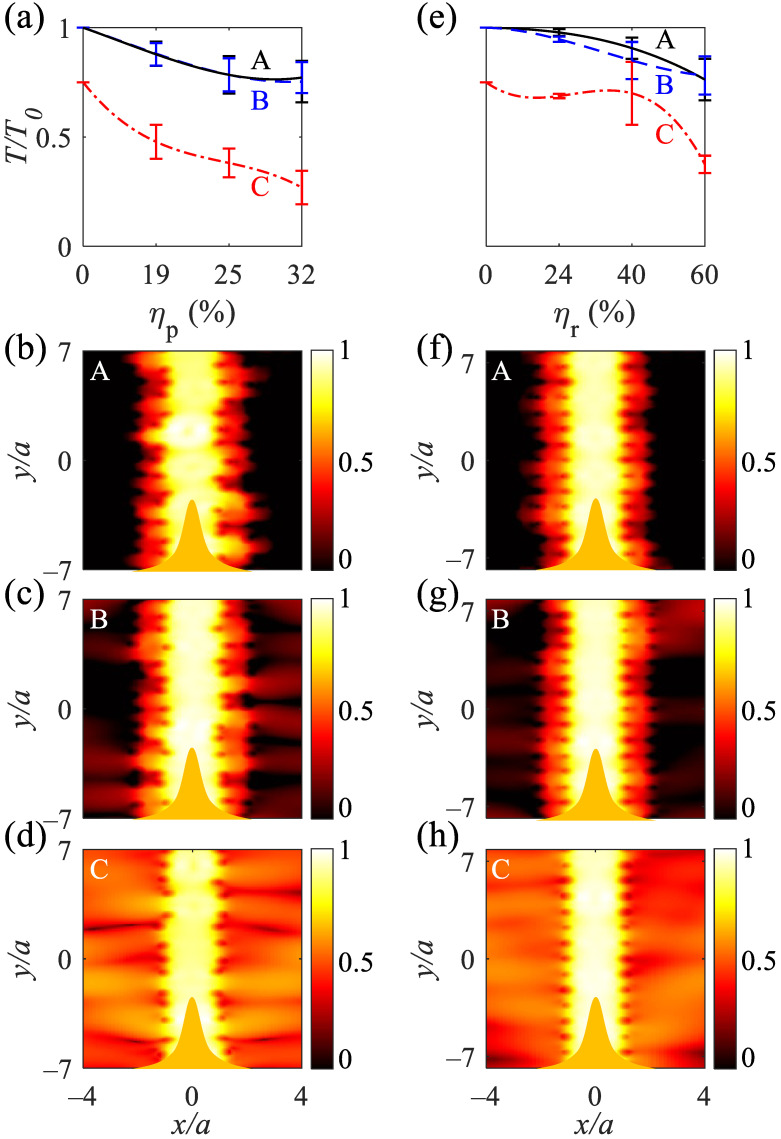
Normalized transmission for the straight A-, B-, and C-type waveguides under the (**a**) position disordering at $\eta_{p}$ = 19, 25, and 32% and (**e**) radius disordering at $\eta_{r}$ = 24, 40, and 60%. (**b**–**d**) Normalized logarithmic electromagnetic power distributions at λ=490 nm for A, B, and C-type waveguides under $\eta_{p}=19\%$. (**f**–**h**) Normalized logarithmic electromagnetic power distributions at λ=490 nm for A, B, and C-type waveguides under $\eta_{r}=24\%$. The orange bell shapes at the bottom of (**b**–**d**,**f**–**h**) show the TM polarized Gaussian waves, propagating from −y to *y*, and are recorded at the top of the waveguides.

**Figure 7 nanomaterials-12-02132-f007:**
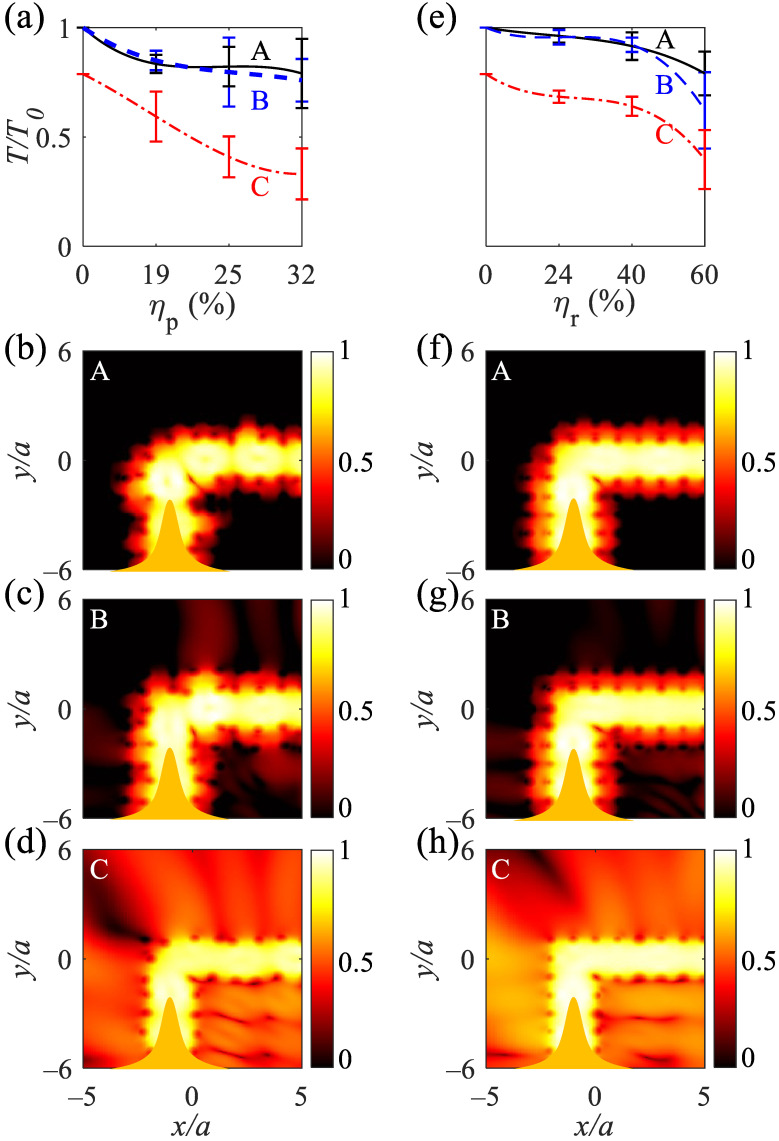
Normalized transmission for the A-, B-, and C-type L-shaped waveguides under the (**a**) position disordering under $\eta_{p}$ = 19, 25, and 32% and (**e**) radius disordering under $\eta_{r}$ = 24, 40, and 60%. (**b**–**d**) Normalized logarithmic electromagnetic power distributions at λ=490 nm for A, B, and C-type L-shaped waveguides under $\eta_{p}=19\%$. (**f**–**h**) Normalized logarithmic electromagnetic power distributions at λ=490 nm for A, B, and C-type L-shaped waveguides under $\eta_{r}=24\%$. The orange bell shapes at the bottom of (**b**–**d**,**f**–**h**) show the TM polarized Gaussian waves, propagating from −y to *y*, and are recorded at the right of the waveguides.

**Figure 8 nanomaterials-12-02132-f008:**
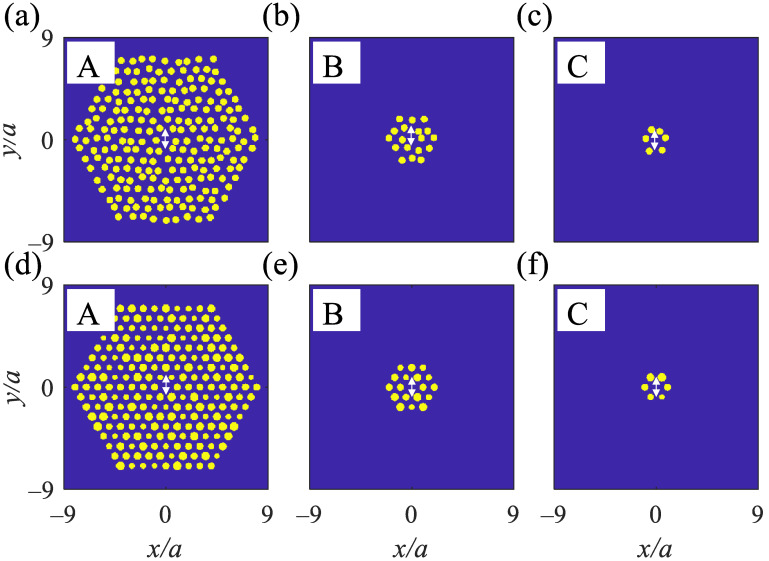
2D MPHC structure under $\eta_{p}=19\%$ contains Ag rods in air with a removed rod at the center surrounded by (**a**) eight, (**b**) two, and (**c**) one rings of Ag rods. The structure under $\eta_{r}=24\%$ containing Ag rods in air with a removed rod at the center surrounded by (**d**) eight, (**e**) two, and (**f**) one rings of Ag rods. Yellow: Ag rods and blue: air. White-double arrows at the center of each cavity show TM polarized dipole sources.

**Figure 9 nanomaterials-12-02132-f009:**
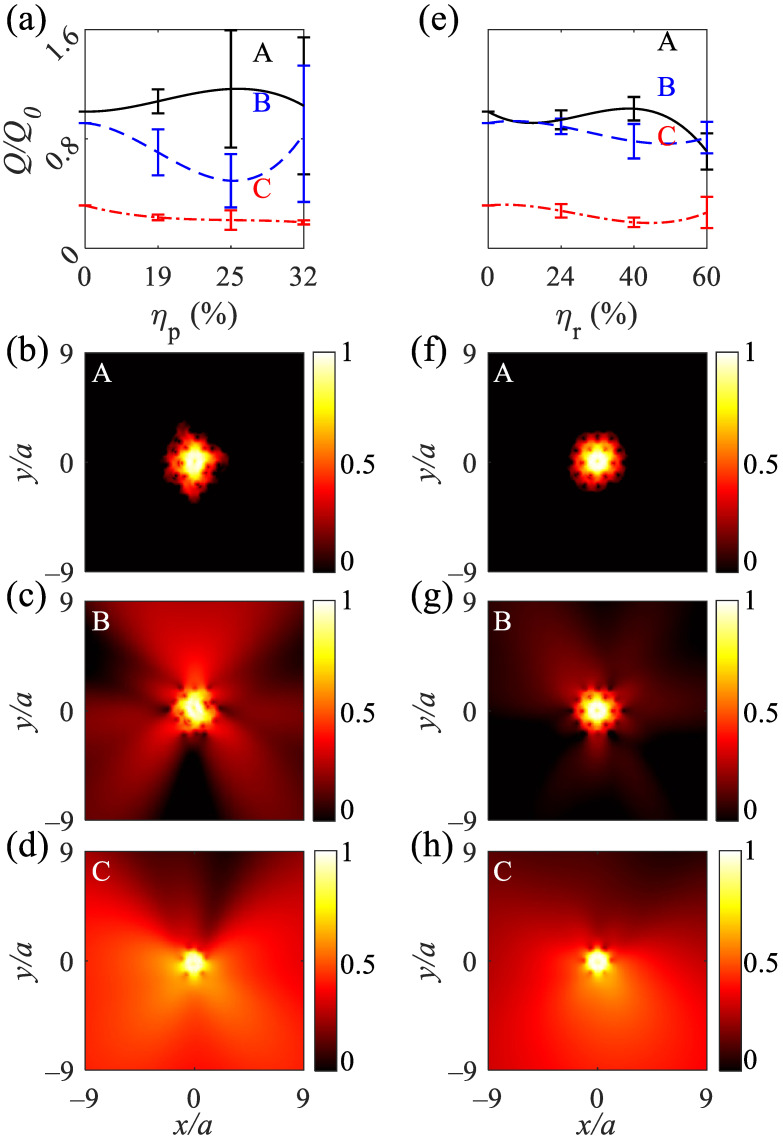
Normalized quality factor for A-, B-, and C-type cavities under (**a**) $\eta_{p}$ = 19, 25, and 32% and (**e**) $\eta_{r}$ = 24, 40, and 60%. (**b**–**d**) Normalized logarithmic electromagnetic power distributions for the A, B, and C-type cavities at λ=428,400, and 400 nm, respectively. (**f**–**h**) Normalized logarithmic electromagnetic power distributions for the A, B, and C-type cavities at λ=424,410, and 410 nm, respectively.

**Table 1 nanomaterials-12-02132-t001:** The comparison of the present work and some recent works.

References	Working Wavelength (nm)	Average Absorption (%)	Material
[45]	280–700	90	Tungsten, Silicon Dioxide
[46]	300–700	92.2	Tungsten, Silicon Dioxide
[47]	350–1250	95.3	Tungsten, Silicon Dioxide
[49]	450–650	65	Aluminum, Gallium Arsenide
[50]	500–700	55	Etalon
[48]	200–500	60	Titanium Nitride
Proposed absorber	200–530	95	Gold

**Table 2 nanomaterials-12-02132-t002:** The average normalized transmission for the straight waveguide under radius and position disorderings.

Waveguide Type	$\eta_{p}=0\%$	$\eta_{p}=19\%$	$\eta_{p}=25\%$	$\eta_{p}=32\%$	$\eta_{r}=0\%$	$\eta_{r}=24\%$	$\eta_{r}=40\%$	$\eta_{r}=60\%$
A	1	0.87	0.78	0.77	1	0.98	0.9	0.78
B	1	0.87	0.78	0.75	1	0.94	0.85	0.76
C	0.75	0.47	0.38	0.27	0.75	0.67	0.7	0.38

**Table 3 nanomaterials-12-02132-t003:** The average normalized transmission for the L-shaped waveguide under radius and position disorderings.

Waveguide Type	$\eta_{p}=0\%$	$\eta_{p}=19\%$	$\eta_{p}=25\%$	$\eta_{p}=32\%$	$\eta_{r}=0\%$	$\eta_{r}=24\%$	$\eta_{r}=40\%$	$\eta_{r}=60\%$
A	1	0.84	0.82	0.8	1	0.96	0.92	0.79
B	1	0.83	0.8	0.79	1	0.96	0.92	0.62
C	0.79	0.59	0.41	0.33	0.79	0.68	0.64	0.4

**Table 4 nanomaterials-12-02132-t004:** The average normalized quality factor for the cavity under radius and position disorderings.

Cavity Type	$\eta_{p}=0\%$	$\eta_{p}=19\%$	$\eta_{p}=25\%$	$\eta_{p}=32\%$	$\eta_{r}=0\%$	$\eta_{r}=24\%$	$\eta_{r}=40\%$	$\eta_{r}=60\%$
A	1	1.1	1.17	1	1	0.9	1.1	0.7
B	0.92	0.7	0.5	0.84	0.92	0.89	0.78	0.8
C	0.3	0.23	0.2	0.19	0.3	0.28	0.19	0.26

## Data Availability

The data presented in this study are available in this article and its Appendix A.

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
