# Peer review of "Towards Perfect Ultra-Broadband Absorbers, Ultra-Narrow Waveguides, and Ultra-Small Cavities at Optical Frequencies"

_nanomaterials, 2022, doi:10.3390/nano12132132_

Round 1

Reviewer 1 Report

The authors proposed a perfect ultra-broadband absorbers, ultra-narrow
waveguides and ultra-small cavities at optical frequencies. The paper is well organized. The results support the conclusions. I think it can be accepted after minor revisions.

1. The background of the manuscript should be improved according to the latest literatures. And there are several literatures are not too concerned about the work.

2. The method used in this work should be explained detailedly. 

3. The results should be compared with others' results.

4. The quality of Figure 4 should be improved. It is not too clear.

5.  The language should be polished.

Author Response

Dear Reviewer A,

Thank you for the valuable comments. We believe that we have answered all comments and applied them to the paper in green color.

The best,

Kiyanoush Goudarzi

Reviewer 2 Report

The paper under consideration has the potential to be further considered for publication, however, there are several major issues that make it difficult to fully understand the paper. I have the following suggestions to be implemented before the final decision on the manuscript.

1)      The presentation of the paper is quite poor. I suggest merging figure 1 and figure 2. In this way, the reader will find it easy to understand the Absorption graphs corresponding to the variations in the parameter of the absorber such as changes in radius and changes in position.

2)      Why is TM polarized light not considered for the absorber?

3)      It would be nice if the wavelengths are represented in nanometers instead of microns.

4)      Figure 2c and figure 2d are not explained in the manuscript.

5)      In figure 1, I am not sure if the author has considered the substrate and its refractive index?

6)      In figure 1, how light is excited on the samples? In-plane or out-of-plane coupling? Draw proper labels in the figure.

7)      Why this range of absorption has been selected for the absorber? It isn’t fully visible? It cant be used in solar cells to enhance their efficiency. How do change the wavelength range of the absorption spectrum?

8)      What is the effect of angle of incidence on the absorption spectrum? This effect hasn’t been explored.

9)      Why there is no E-field or H-field distribution figure for the absorber structure?

10)   I am not sure how absorption has been achieved, because it requires some sort of cavity or resonators where light is confined such as MIM structures or hybrid structures like this https://doi.org/10.1016/j.optmat.2021.111906? Please explain related to the reference I provided. 

11)   A performance comparison table is required for the type A, B, and C waveguides studied in the paper.  It is not easy to understand the figures.

12)   I suggest adding the E-field distribution of all three types of waveguides at the optimized geometry. Because in the paper the E-field distribution for only one type of waveguide is shown.

13)   The author has to draw a figure for the change in position and radius for the straight and bend waveguide. At this moment, it is not easy to understand what the waveguide geometry looks like with those modifications. Also, do it for the cavity design.

14)   Conclusion section has to be modified. It does not give any information on the results obtained. I suggest removing unnecessary things and writing the main results obtained in the paper such as Q-factor, Absorption percentage, wavelength range etc.

15)   Abstract section is also confusing. First of all, it should be small and provide only the main results obtained in the paper. 

Author Response

Dear Reviewer B,

Thank you for the valuable comments. We believe that we have answered all comments and applied them to the paper in yellow color.

The best,

Kiyanoush Goudarzi

Round 2

Reviewer 2 Report

I suggest the author to add the E field distribution of the perfect absorber.

And make corrections in the caption of figure 1. Its a and d, not a and b.

Author Response

Dear Reviewer B,

Thank you for the valuable comments and suggestions. We answered the comments and suggestions and applied them to the paper.

The best,

Kiyanoush Goudarzi
